# Pre-Processing Method for Contouring the Uptake Levels of [18F] FDG for Enhanced Specificity of PET Imaging of Solitary Hypermetabolic Pulmonary Nodules

**DOI:** 10.3390/jcm10071430

**Published:** 2021-04-01

**Authors:** Piotr Szumowski, Artur Szklarzewski, Łukasz Żukowski, Saeid Abdelrazek, Małgorzata Mojsak, Katarzyna Porębska, Ewa Sierko, Janusz Myśliwiec

**Affiliations:** 1Department of Nuclear Medicine, Medical University of Bialystok, M. Skłodowskiej-Curie St. 24A, 15-276 Bialystok, Poland; lukzuk85@wp.pl (Ł.Ż.); saeid@op.pl (S.A.); mni@o2.pl (M.M.); janusz.mysliwiec69@gmail.com (J.M.); 2Department of Nuclear Medicine, Comprehensive Cancer Center of Białystok, 15-027 Bialystok, Poland; kporebska@onkologia.bialystok.pl; 3Voxel Medical Diagnostic Centres, VOXEL S.A., 30-663 Kraków, Poland; 4Strictsense.com, Zachodnia, Number 2G/62, 15-345 Białystok, Podlaskie, Poland; artur.szklarzewski@gmail.com; 5Department of Radiotherapy, Comprehensive Cancer Center of Białystok, 15-027 Bialystok, Poland; ewa.sierko@iq.pl

**Keywords:** hypermetabolic pulmonary nodule, PET/CT, lung cancer, FDG

## Abstract

Background: The paper presents a pre-processing method which, based on positron-emission tomography (PET) images of ^18^F-fluorodeoxyglucose ([18F] FDG) hypermetabolic pulmonary nodules, makes it possible to obtain additional visual characteristics and use them to enhance the specificity of imaging. Material and Methods: A retrospective analysis of 69 FDG-PET/CT scans of solitary hypermetabolic pulmonary nodules (40 cases of lung cancer and 29 benign tumours), where in each case, the standardised uptake value of the hottest voxel within the defined volume of interest was greater than 2.5 (SUVmax > 2.5). No diagnosis could be made based on these SUVmax values. All of the PET DICOM images were transformed by means of the pre-processing method for contouring the uptake levels of [18F] FDG (PCUL-FDG). Next, a multidimensional comparative analysis was conducted using a synthetic variable obtained by calculating the similarities based on the generalised distance measure for non-metric scaling (GDM2) from the pattern object. The calculations were performed with the use of the R language. Results: The PCUL-FDG method revealed 73.9% hypermetabolic nodules definitively diagnosed as either benign or malignant lesions. As for the other 26.1% of the nodules, there was uncertainty regarding their classification (some had features suggesting malignancy, while the characteristics of others made it impossible to confirm malignancy with a high degree of certainty). Conclusions: Application of the PCUL-FDG method enhances the specificity of PET in imaging solitary hypermetabolic pulmonary nodules. Images obtained using the PCUL-FDG method can serve as point of departure for automatic analysis of PET data based on convolutional neural networks.

## 1. Introduction

The strategy for assessing the characteristics of a nodule on the basis of positron-emission tomography (PET) with ^18^F-fluorodeoxyglucose ([18F] FDG) relies on using the method’s capacity to monitor the metabolic processes occurring in lesions. The levels of [18F] FDG uptake in a nodule (measured with standardised uptake value—SUV) depends mainly on the number of viable neoplastic cells, the type of neoplastic cells, and the size of lesion, but also on the number of inflammatory cells, among others. Therefore, it is very difficult to establish a precise cut-off point for the SUV parameter that would unequivocally indicate malignancy (SUV = 2.5 is usually taken as the cut-off value) [1]. Nevertheless, the method has proved to be a very good predictor of malignancy in solitary pulmonary nodules, where specificity usually reaches 80% (in the case of CT, it normally does not exceed 60%) and sensitivity can be as high as 90–95%. The method’s lower specificity is primarily associated with the possibility of false positive results (hypermetabolic nodules can also be caused by pneumonia, sarcoidosis, tuberculosis, or amyloidosis) or, less frequently, false negative ones (small lesions of less than 1 cm in diameter, follicular–bronchiolar carcinomas, or neuroendocrine tumours) [2,3,4]. 

The dynamic development of information technologies has extended the scope of computer-based image processing and analysis, even in medicine. Radiomics, which is concerned with the identification, extraction, and analysis of image features [5], offers a variety of methods that can be used in the imaging of lung nodules. PET, among others, when analysed in terms of radiomic features, can provide a solid foundation for assessment and classification of pulmonary nodules [6,7]. 

This paper presents a pre-processing method, which, based on FDG-PET of hypermetabolic pulmonary nodules, makes it possible to reveal additional visual characteristics that can be used for enhancing the specificity of imaging in diagnosis of nodules. This method can become an alternative to the approach relying merely on quantitative analysis of SUV. 

## 2. Materials and Methods

### 2.1. Patients 

This study involves a retrospective analysis of 69 FDG-PET/CT scans showing solitary hypermetabolic pulmonary nodules, including 40 cases of pathologically confirmed lung cancer (glandular and squamous cell carcinomas). The other 29 images show benign nodules. Their benign character was confirmed by the dynamics of the morphological changes in the nodules in subsequent radiological imaging (i.e., disappearance of the lesion and reduction or stabilisation of tumour size in the period of 2 years without anticancer interventions) or by transthoracic biopsy [8]. 

It must be added that a solitary nodule is, as the widely accepted definition claims, a well-demarcated, circular or oval condensation of the pulmonary parenchyma revealed in a PET/CT scan, equal to or smaller than 30 mm, and found in an originally unchanged lung [9]. Because of the resolution of PET, a minimum size of nodule (≥10mm) was an additional limitation to selecting PET images for analysis. 

### 2.2. FDG-PET/CT Imaging—Data Acquisition

The PET/CT images were taken using a Discovery 610 scanner (GE Medical Systems, Waukesha, WI, USA). Before being intravenously injected with [18F] FDG radiotracer, patients had been fasting for 6 h, and their blood glucose was <7.8 mmol/L (140 mg/dL). The scan spanned the base of the skull to one-third of the upper part of the femur (about 5–7 bed, where emission acquisition duration per bed position equals 2 min·bed−1), 60 min after an intravenous injection with 0.9 mCi/kg of [18F] FDG. 

CT scans were performed with the following parameters: current, 120–170 mA; voltage, 120 kV; slice thickness, 3.33 mm; and reconstruction interval, 3.33 mm. Afterward, CT-based attenuation-corrected PET images were acquired in three-dimensional mode with two minutes per bed position and were reconstructed using an iterative algorithm with a 192 × 192 matrix. In addition, a non-contrast CT scan targeted to the lung lesion with a slice thickness of 1.25 mm was also obtained for each patient. PET metabolic parameters were automatically generated by PETVCAR, a semiquantitative software embedded in GE workstation. SUVs were calculated according to the following formula: SUV = radioactivity concentration (mCi/kg)/(injected activity (mCi)/body weight (kg)). More precisely, in our project, we used SUVmax (SUV of the hottest voxel within a defined volume of interest (VOI)) because it is the most widely used parameter, easy to use, and operator-independent [10]. We assumed in this paper that the SUVmax value, indicating a hypermetabolic nodule, would be ≥2.5.

PET-CT scanners produce separate PET and CT images in formats specified by the Digital Imaging and Communications in Medicine (DICOM) standards. In further stages, information obtained from non-contrast CT (e.g., evaluation of tumour margins and presence of subsolid lesions or calcifications) was omitted.

### 2.3. Image Pre-Processing

Using the pydicom programming library (P1), patient records from the DICOMDIR file were read. After data anonymisation, PET series (DICOM files with a layer thickness of 3.3 mm) for every patient were processed. We analysed axial projection PET DICOM images containing only the lungs. Using opencv (P2) and skimage (P3) programming libraries, the following pipeline was applied for every image:-Read [18F] FDG uptake values from DICOM pixel data;-Resampled from original space to new one using <1, 1>;-Obtained 700 × 700 matrix was normalised to a value range <0, 1>;-An OR bitwise operation with an all-ones matrix was performed;-Obtained values used as the transparency channel were superimposed against a white background and saved as a grayscale PNG image.

As a result, a contour plot of the image was obtained. The entire operation was named a pre-processing method for contouring the uptake levels of [18F] FDG (PCUL-FDG). Figure 1 shows a sample result of the method’s application. 

As a result of the transformation, the area of increased [18F] FDG uptake in the pulmonary nodule was converted into a clear, graphic structure, which was described using the following terms: 

[18F] FDG uptake contour—the line that divides areas of significantly differing levels of [18F] FDG uptake. It is usually asymmetrical and helical or oval in shape. 

Maximum [18F] FDG uptake (maximum emission)—the area where the highest [18F] FDG uptake values were recorded; a small-sized area, noticeable as a white dot surrounded with a black band, usually inside the most nested contour, with the exception of contours that are not nested individually and where, within a given contour, there are many areas of other nested or non-nested contours (Figure 2)*—*in such a case, maximum uptake may not be located in the most nested contour.

### 2.4. Feature Extraction

Below are examples of PET image sequences obtained using the PCUL-FDG method depicting a benign pulmonary nodule (Figure 2) and a malignant pulmonary nodule (Figure 3). 

For the purposes of the method, in order to enable a coherent description of the transformed PET sequences (see Figure 2 and Figure 3), we defined the features that described the graphic structures visible in the area of the pulmonary nodules. With those visual features, we could obtain information that was useful in assessing the characteristics of the lesions. 

### 2.5. Degree of Nesting of [18F] FDG Uptake Contours (Feature I)

Feature 4 refers to the number of contours nested one within another in the pulmonary nodule. The highest degree of nesting present in the entire PET sequence is taken as the value of the feature. This is a visual representation of the degree of metabolic variability of an analysed pulmonary nodule. Figure 4 represents nesting of contours that is typical of various PET images.

Figure 5 show slightly different kinds of nesting, where inside a major contour we can observe many smaller nested or non-nested contours. 

### 2.6. Change in the Shape of [18F] FDG Uptake Contours (Feature II) 

Feature II refers to the extent to which the shape of [18F] FDG uptake contours is altered in the subsequent sequences of transformed PET images of pulmonary nodules. Figure 5, Figure 6 and Figure 7 present different stages of alterations in the shape of these contours.

### 2.7. Shifts in the Area of Maximum [18F] FDG Uptake within Nodule Borders (Feature III) 

Feature III refers to the change in the location of the area with maximum [18F] FDG uptake within the contour that surrounds it in subsequent scans of transformed PET images of pulmonary nodules. Figure 8, Figure 9 and Figure 10 show examples of such shifts in the location of maximum uptake. 

### 2.8. Size of the Contour Encompassing the Area of Maximum [18F] FDG Uptake (Feature IV)

Feature IV refers to the largest size of the contour that directly surrounds maximum [18F] FDG uptake, as observed in a sequence of PET images of a pulmonary nodule. Examples can be seen in Figure 11.

### 2.9. Analysis of Features 

Each image of a pulmonary nodule transformed by means of PCUL-FDG was described in terms of the four above-mentioned features, which are referred to as diagnostic variables for the purposes of this analysis. The values of the diagnostic variables were determined based on weighted ordinal scales. The weights were determined a priori on the basis of preliminary analysis of the values of particular features with respect to their informative richness. As can be seen in Figure 12, variables I and II have widely varying values within both studied groups. Therefore, it can be assumed that they have a higher informative potential when comparing transformed images. 

Eventually, the following criteria for evaluating PET scans were adopted in the PCUL-FDG approach: 

Degree of contours nesting (variable no. 1);

Scale: 1—no nesting, 2—low degree of nesting, 3—moderate degree of nesting, 4—high degree of nesting;

Weight: 0.6;

Type: stimulant;

Change in the shape of contours (variable no. 2);

Scale: 1—slight, 2—moderate, 3—considerable;

Weight: 0.3;

Type: stimulant;

Shifts in the area of maximum [18F] FDG uptake (variable no. 3);

Scale: 1—slight shift in the location of maximum uptake, 2—distinct shift, 3—considerable shift;

Weight: 0.05;

Type: stimulant;

Size of the contour encompassing maximum [18F] FDG uptake (variable no. 4);

Scale: 1—small size, 2—medium size, 3—large size;

Weight: 0.05;

Type: neutral variable.

Next, a multivariate comparative analysis was performed. Its purpose was to order the PET scans of hypermetabolic pulmonary nodules according to the degree of malignancy, relying on a synthetic variable obtained by calculating similarities based on GDM2 (generalised distance measure for non-metric scaling) from the pattern object. The calculations were performed using R language (function: dist.GDM2 in the clusterSim package) [11]. 

## 3. Results 

Figure 13 contains calculated values of a synthetic variable based on GDM2. PET scans were sorted according to their distance from the pattern object. 

Additionally, borders dividing the particular categories of lesions, determined by means of the maximum gradient method [6], were superimposed over the linearly ordered PET images. Three areas of the highest absolute value of the differences between subsequent values of the synthetic variable were identified. On this basis, three limit values of the synthetic variable (distance from the pattern object) could be established for the set of PET images in question. They divide the analysed pulmonary nodules into the following groups:Group I—distance from pattern object in the range of 0–0.29—lesions with distinct, prevalent features of a malignant tumour (33 cases of malignant tumours, i.e., 82.5% of all the cases of malignant tumours under consideration);Group II—distance from pattern object in the range of 0.29–0.4—lesions with features indicative of malignancy (6 cases of malignant tumours and 1 case of a benign lesion, which account for 15% of all the malignant lesions and 3.5% of the benign lesions, respectively);Group III—distance from pattern object in the range of 0.4–0.54—lesions the values of whose features made it impossible to confirm malignancy with a high degree of certainty (1 malignant tumour, 10 benign lesions, i.e., 2.5% of the analysed malignant tumours and 34.5% of analysed benign lesions, respectively);group IV—distance from pattern object in the range of 0.54–1—lesions with features suggesting benignity (18 cases of the benign lesions, i.e., 62% of all such lesions under consideration).

The number of pulmonary nodules from the particular categories in each of the defined groups allows us to determine that 0.4 is the value of the distance from the pattern object above which a lesion can be expected to be benign. If the figure is lower than or equal to 0.4, it can be expected that the lesion’s features will clearly indicate malignancy.

## 4. Discussion

The use of the PCUL-FDG method in the present study significantly contributed to enhancing the specificity of PET imaging in evaluating the characteristics of solitary hyperbolic pulmonary nodules.

Based on SUVmax alone, as has been done heretofore, does not enable unequivocal diagnosis, as the studied groups of malignant and benign hypermetabolic pulmonary nodules do not differ sufficiently. This is associated with the condition that we assumed in our methodology that a hypermetabolic nodule (whether malignant or benign) had to have SUV ≥ 2.5, which follows from the generally accepted risk stratification of malignancy in pulmonary nodules. Nodules with SUV < 2.5 are lesions with low [18F] FDG uptake and with very low risk of potential malignancy. Nodules with SUV ≥ 2.5, so-called hypermetabolic ones, have a much higher diagnostic uncertainty when assessed on the basis of PET scans [1].

The PCUL-FDG method allowed us to identify as many as 51 (73.9%) nodules with a confident diagnosis of benignity or malignancy (18 benign and 33 malignant ones) out of 69 hypermetabolic pulmonary nodules. As far as the other 18 (26.1%) nodules are concerned, they could not be confidently evaluated. The present method divides them into two groups: nodules whose features assumed values suggesting malignancy (7 tumours:6 malignant ones and 1 benign one) and nodules whose features assumed values that did not make it possible to confirm malignancy with a high degree of certainty (11 tumours:10 benign ones and 1 malignant one). To simplify, it can be assumed that if GDM2 is ≤0.4, distinct features of malignancy can be expected in a hyperbolic pulmonary nodule. Given such assumptions, the accuracy of diagnosis reaches 97.5% (39 out of 40 truly positive cases under analysis) (Figure 13).

It should be emphasised, however, that the presented numerical values describing the effects of applying the PCUL-FDG method concern a specific set of scans. It would be advisable to conduct a similar study on a larger sample of cases, also including images obtained from other PET/CT scanners. This would make it possible to assess the repeatability of the method.

The observed respiratory mobility of the nodules, particularly those located in the lower sections of the lungs [12,13], had little impact on the evaluation of features obtained after transforming images by means of the PCUL-FDG method. The discussed method does seem to be slightly affected by this kind of measurement failure because should it occur, the entire structure, after transformation, is moved by a certain vector. The structure itself, however, does not become distorted.

However, one should be aware that artifacts resulting from respiratory mobility are a highly diverse phenomenon, both in terms of intensity and its impact on the accuracy of the measurement. In the analysed PET images, the respiratory mobility was not observed in a degree allowing to clearly state the resistance of the PCUL-FDG method to this type of measurement defect. This issue may be the subject of further analysis.

In our method, application of appropriate transformation of the PET images of hypermetabolic pulmonary nodules enabled us to evaluate its character on the basis of visual features that are not discernible in the original images. This approach of relying on extraction of features from a primary image is typical of radiomics, where image transformation techniques are used for advanced analysis, which can potentially reveal additional disease features and facilitate classification of neoplastic lesions. Therefore, in the context of radiomics, the PCUL-FDG method can be treated as this type of analysis, using the characteristics of shapes to assess the character of pulmonary nodules [14,15,16,17].

The contour image obtained through the PCUL-FDG method is characteristic enough (so-called helical shapes of the contours and their properties) to be suitable for application in hybrid automatic detection systems of malignant lesions, i.e., PET/CT and PET/MR, in the hope of achieving additional clinical benefits resulting from the use of multiparameter models [18]. Nevertheless, with the PCUL-FDG method, the analysis of PET images of pulmonary nodules alone revealed significant differences between the benign and malignant lesions. This is particularly noticeable in Figure 2 and Figure 3, where the analysed nodules differ in terms of the degree of nesting and the shape of the contours of [18F] FDG uptake, as well as with regard to the change in the location of maximum [18F] FDG uptake. The change in the shape of contours is more pronounced, and a higher degree of nesting can be observed. This suggests a differentiation and intensity of the changes in the value of FDG uptake in a given area. Moreover, in the case of a benign lesion, the site of the maximum [18F] FDG uptake does not shift abruptly, whereas in a malignant lesion, significant shifts in its location can be noticed.

Therefore, it seems worthwhile to develop the terminology related to the features that describe PET images transformed by means of the PCUL-FDG method. Fine-tuning the definitions, refining their interpretations, and introducing a description based on topological characteristics of the shapes should have an impact on the practical aspect of the presented approach and inspire the development of software that would enable using the method on a larger scale.

## 5. Conclusions

The results obtained using the PCUL-FDG method are promising. The applied transformation of PET images of hypermetabolic pulmonary nodules makes it possible to evaluate their characteristics on the basis of the visual features of the yielded contour structure.

The conducted analysis of linear ordering with the use of a synthetic variable calculated on the basis of GDM2 (Figure 13) enables an evaluation of a given pulmonary nodule in terms of features indicating its malignancy.

The images obtained through the PCUL-FDG method are characteristic enough to be a point of departure for automated analysis of PET images on the basis of topological features or with the use of, e.g., convolutional neural networks.

## Figures and Tables

**Figure 1 jcm-10-01430-f001:**
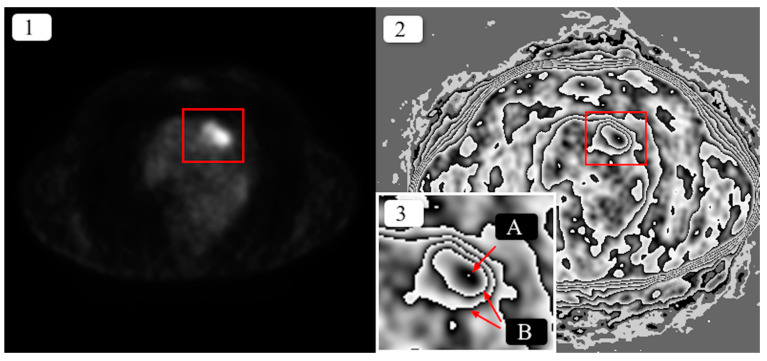
Result of the transformation of a single positron-emission tomography (PET) DICOM image of the lung using the pre-processing method for contouring the uptake levels of ^18^F-fluorodeoxyglucose ([18F] FDG) (PCUL-FDG) method. In both versions of the axial projection image, the red rectangle marks an area with increased [18F] FDG uptake in a pulmonary nodule within a 3.3-mm-thick layer. (**1**) Original image, (**2**) transformed image, (**3**) close-up of the section marked with red, where A: maximum [18F] FDG uptake, B: nested uptake contours.

**Figure 2 jcm-10-01430-f002:**
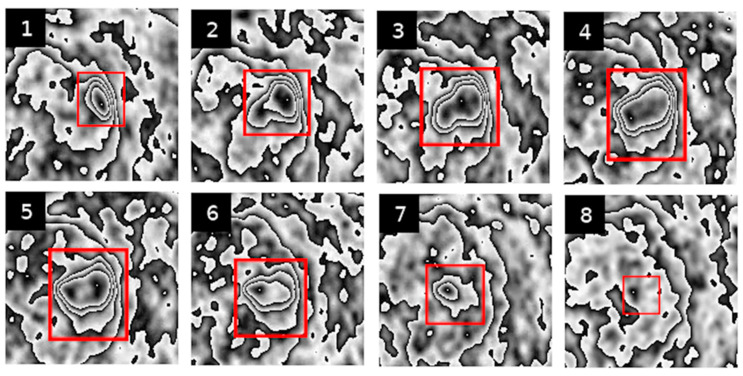
Result of the transformation of PET image sequences using the PCUL-FDG method. The red square indicates the site of a malignant hypermetabolic pulmonary nodule (Figures 1–6 show successive slices of a lung nodule).

**Figure 3 jcm-10-01430-f003:**
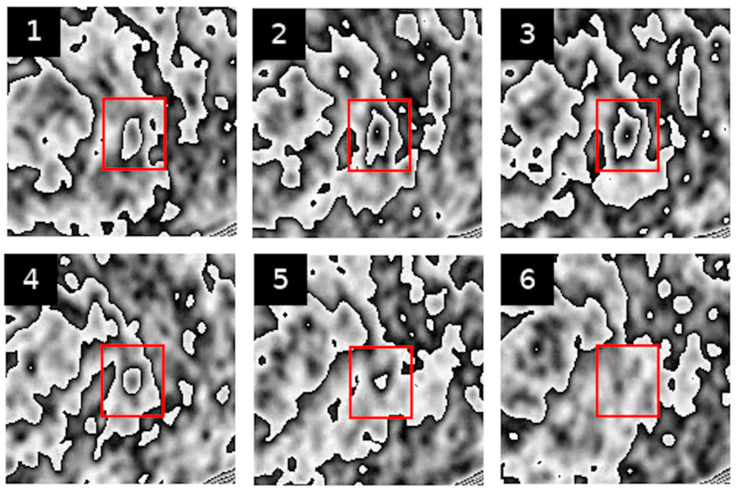
Result of the transformation of PET image sequences using the PCUL-FDG method. The red square indicates the site of a benign hypermetabolic pulmonary nodule (Figures 1–6 show successive slices of a lung nodule).

**Figure 4 jcm-10-01430-f004:**
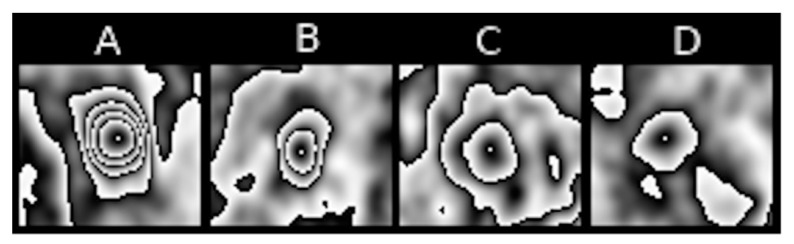
Examples of [18F] FDG uptake contour nestings within various types of pulmonary nodules: (**A**) high degree, (**B**) moderate degree, (**C**) low degree, (**D**) none.

**Figure 5 jcm-10-01430-f005:**
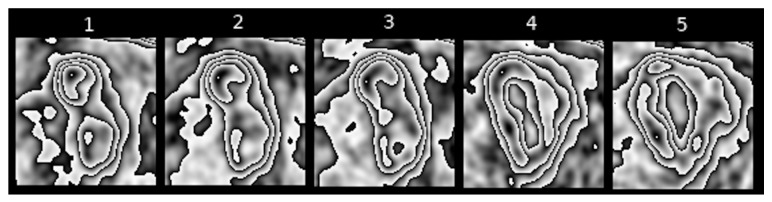
Example of a sequence of transformed PET images of a pulmonary nodule with considerably altered shape of [18F] FDG uptake contours.

**Figure 6 jcm-10-01430-f006:**
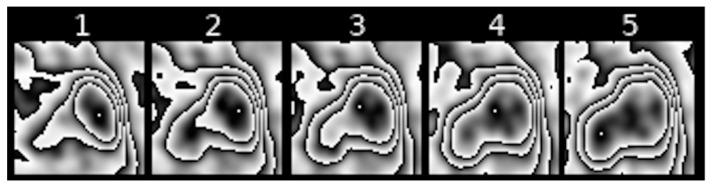
Example of a sequence of transformed PET images of a pulmonary nodule with moderately altered shape of [18F] FDG uptake contours.

**Figure 7 jcm-10-01430-f007:**
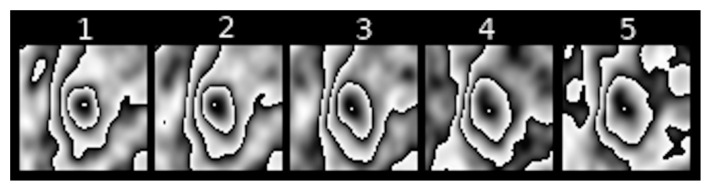
Example of a sequence of transformed PET images of a pulmonary nodule with slightly altered shape of [18F] FDG uptake contours.

**Figure 8 jcm-10-01430-f008:**
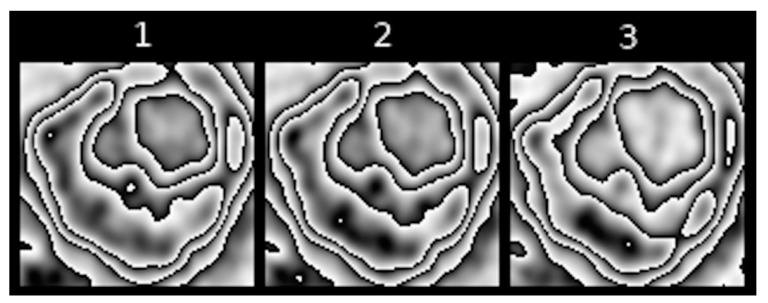
Example of a considerable shift in the location of maximum [18F] FDG uptake in a sequence of PET images of a pulmonary nodule.

**Figure 9 jcm-10-01430-f009:**
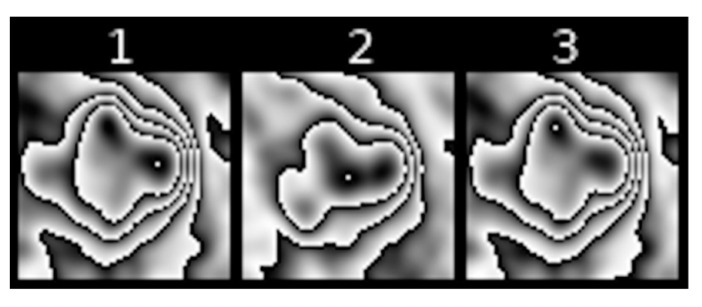
Example of a distinct shift in the location of maximum [18F] FDG uptake in a sequence of PET images of a pulmonary nodule.

**Figure 10 jcm-10-01430-f010:**
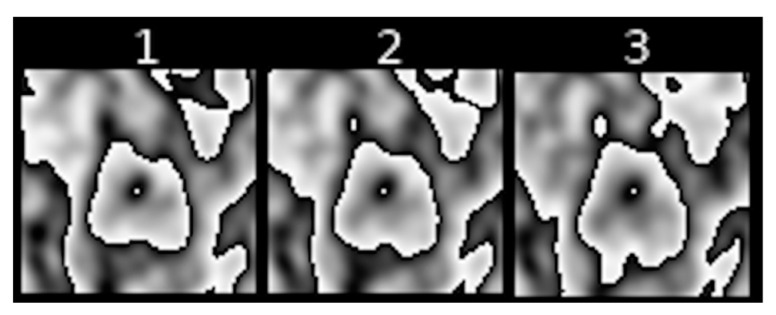
Example of a slight shift in the location of maximum [18F] FDG uptake in a sequence of PET images of a pulmonary nodule.

**Figure 11 jcm-10-01430-f011:**
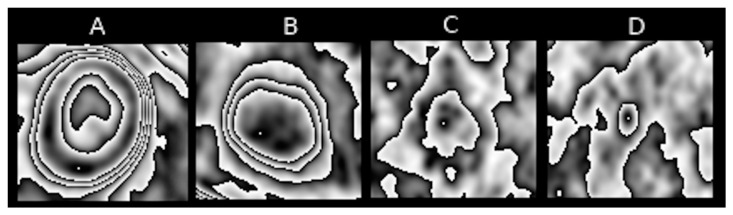
Examples of different sizes of contours encompassing maximum [18F] FDG uptake in four different pulmonary nodules. (**A**) Large size, (**B**,**C**) medium size, (**D**) small size.

**Figure 12 jcm-10-01430-f012:**
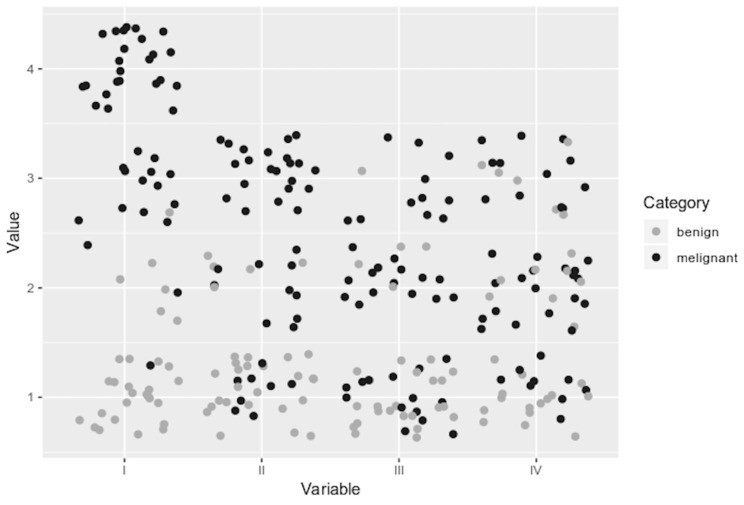
Distribution of the values of analysed variables. For better clarity, a jittered points plot was used.

**Figure 13 jcm-10-01430-f013:**
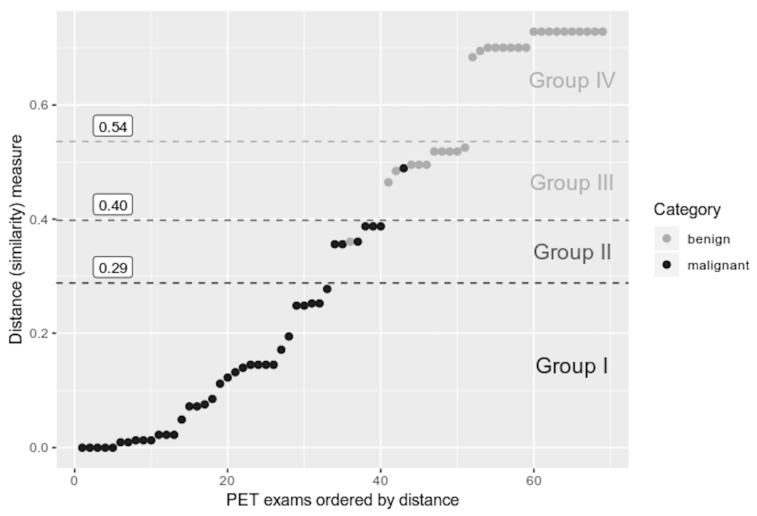
Values of synthetic variable based on GDM2 for each analysed hypermetabolic nodule in PET images.

## Data Availability

The datasets used and/or analysed during the current study are available from the corresponding author on reasonable request.

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
