# Peer review of "Pre-Processing Method for Contouring the Uptake Levels of [18F] FDG for Enhanced Specificity of PET Imaging of Solitary Hypermetabolic Pulmonary Nodules"

_jcm, 2021, doi:10.3390/jcm10071430_

Round 1
Reviewer 1 Report
I have a few comments to the authors of the article:
- I propose to use the currently adopted nomenclature of names of radiopharmaceuticals. Abbreviation should be used: [18F] FDG
- I propose to write in how many patients the diagnosis was verified on the basis of the hist-path examination
- I propose to use only SI units. lines: 77.79. The dose of 0.9 mCi / kg multiplied by 70 kg is 63 mCi. Is this dose too high, however?
- Have the authors performed image analysis using different image reconstruction / parameter protocols (do the results depend e.g. on the number of iterations, other parameters?)
- I propose a more detailed explanation why the analysis does not depend on motion artifacts. Movement artifacts blur the image and should disrupt its structure. Did the authors compare the analysis of the classical exam and the gated exam?
- Line 221 - is there a typo in this line?
- I think the article is interesting and should be published. The article presents original observations. I believe that the described method requires further development..
Author Response
I wish to express my profound gratitude for Your valuable comments concerning my work.
With reference to the comments, below I present the responses to them:
“I propose to use the currently adopted nomenclature of names of radiopharmaceuticals. Abbreviation should be used: [18F] FDG”
- As suggested abbreviations "[18F] FDG" were introduced in the manuscript
“I propose to write in how many patients the diagnosis was verified on the basis of the hist-path examination”
- Information about the number of patients whose diagnosis were verified on the basis of the hist-path examination is included in the lines 63-64 of the manuscript. It is stated there that in the case of 40 patients lung cancer was confirmed pathologically (glandular and squamous cell carcinomas). As to the remaining number of 29 patients, images showed benign nodules and there was no need to verify them pathologically due to the reasons specified in the manuscript.
“I propose to use only SI units. lines: 77.79. The dose of 0.9 mCi / kg multiplied by 70 kg is 63 mCi. Is this dose too high, however?”
- minimal acceptable administered activity of FDG was calculated according to EANM guidelines
- For systems that apply a PET bed overlap of >30 %, the minimum FDG administered activity is calculated as follows:
FDG (MBq)=7 (MBq·min·bed−1·kg−1) × patient weight (kg)/emission acquisition duration per bed position (min·bed−1).
If emission acquisition duration per bed position equals 2min·bed−1 then
FDG(MBq)= 7 (MBq·min·bed−1·kg−1) × patient weight (kg)/2 (min·bed−1), when shortened, it will be as it is in our work:
FDG(MBq)= 3.5 (MBq) × patient weight (kg) or 0.9mCi x patient weight (kg)
- In the section “FDG-PET/CT imaging - data acquisition” of the manuscript (line 79) additional information was introduced as to the time of emission acquisition duration per bed position.
“Have the authors performed image analysis using different image reconstruction / parameter protocols (do the results depend e.g. on the number of iterations, other parameters?)”
- In principle, the PCUL-FDG method should be suitable for wholesale research processing. We wanted to obtain the matrix form of a single DICOM image as soon as possible, which is further processed as an element of a given series.
- This allows for the presentation of contour images for the entire sequence of lung images and their analysis both individually and in a possible relationship with the other images in the sequence. From this point of view, the techniques of image reconstruction allowing the visualization of the entire test / measurement were not the subject of our considerations. Which of course does not change the fact that trying to get a 3D image from contour images is also interesting.
“I propose a more detailed explanation why the analysis does not depend on motion artifacts. Movement artifacts blur the image and should disrupt its structure. Did the authors compare the analysis of the classical exam and the gated exam?”
- We have changed the paragraph on breathing movements. Our previous explanation of the resistance of our method to motion artifacts was indeed too restrained. It should be pointed out that this issue is definitely worth of further in depths analysis.
Line 221 - is there a typo in this line?
- In the line 221 an abbreviation was used such as “GDM2”. There are two types of GDM – one type is GDM1 and the other type is GDM2. We used GDM2 in the manuscript because it is applicable in the ordinal scale.
I would be very grateful if You could undertake the task of evaluating my paper once again in its present form and reconsider for publication the revised version.
Yours sincerely
Piotr Szumowski, M.D.
Reviewer 2 Report
Line 15 formatting
Line 23, 26, 27 et so on relate spacing: double white space
The work involves a pre-processing method that, based on PET images of FDG hypermetabolic pulmonary nodules, allows for additional visual features and uses them to improve imaging specificity.
Unfortunately, it doesn't mention what technical features these images must have.
It also doesn't specifically address how and with what mathematical algorithms these are analyzed.
It doesn't mention whether there are measurement errors evaluated by the propagation of the mathematical processing of the data.
I also notice that
-the absence of the data of the patients analyzed even just a frequency table with the clinical and socio-demographic characteristics of the patients who had the nodule
-a summary table value SUV vs features correlation of the examined nodules.
These two tables would have contributed to the scientificity and the possibility of work correlating with those that in my opinion will follow.
Author Response
I wish to express my profound gratitude for Your valuable comments concerning my work.
With reference to the comments I present below the responses to them:
“Line 15 formatting”
- corrected
“Line 23, 26, 27 et so on relate spacing: double white space”
- corrected
“Unfortunately, it doesn't mention what technical features these images must have”
- We used DICOM images from which FDG values were taken. There are no specific requirements for the input images themselves. It is enough that they meet the DICOM-PET standards.
- We have added a slightly more detailed description of data extraction from DICOM / DICOM DIR and we have described the processing pipeline in a clearer way.
“It also doesn't specifically address how and with what mathematical algorithms these are analyzed”
- We have additionally included the programming libraries used to construct the DICOM image processing pipeline and obtain the final contour image. At this stage of the presentation of the PCU1-FDG method, we wanted to show its characteristic in terms of relatively easy localization of tumors and showing the difference between a benign and a malignant tumor. It is undisputable that the contour image can be analyzed using highly formal mathematical methods, e.g. topology of shapes, but this is an issue for possible future considerations.
“It doesn't mention whether there are measurement errors evaluated by the propagation of the mathematical processing of the data”
- Due to the nature of the measurement scale (ordinal scale) we used for the contour image features, we had to apply the synthetic variable GDM2 for statistical analysis. Any errors in the values of individual features did not turn out to be destructive for the entire calculation.
- Potential shortcomings of the scale used or the measurement of features itself have been "mitigated" by the GDM2 distance measure from the pattern.
- These distance mesure is a contextual measure and uses information about the relationships in which the compared objects remain in relation to other objects.
- Nevertheless, it is highly recommended to check the PCUL-FDG method on a larger number of tests. In this way we are able to detect possible susceptibility of the method to measurement errors.
“the absence of the data of the patients analyzed even just a frequency table with the clinical and socio-demographic characteristics of the patients who had the nodule”
“summary table value SUV vs features correlation of the examined nodules”
“These two tables would have contributed to the scientificity and the possibility of work correlating with those that in my opinion will follow”
- The paper presents a pre-processing method which, basing only on PET images of FDG hypermetabolic pulmonary nodules, makes it possible to obtain additional visual characteristics and uses them to enhance the specificity of imaging. Nevertheless, in the following studies, the combination of images obtained using the PCUL-FDG method with the clinical and socio-demographic characteristics of the patients and SUVs can serve as point of departure for automatic analysis of PET data based on convolutional neural networks.
I would be very grateful if You could undertake the task of evaluating my paper once again in its present form and reconsider for publication the revised version.
Yours sincerely
Piotr Szumowski, M.D.